# Emotions and Media: Emotional Regime and Emotional Factors of Selective Exposure

Jose Manuel Rivera Otero *, Diego Mo-Groba *  and Gemma Vicente Iglesias 

Department of Political Science and Sociology, Universidad de Santiago de Compostela, Praza do Obradoiro, 0, 15705 Santiago de Compostela, A Coruña, Spain; gemma.vicente@rai.usc.es
* Correspondence: josemanuel.rivera@usc.es (J.M.R.O.); diego.mo.groba@usc.es (D.M.G.)

**Abstract:** The central objective of this research is to describe the role of emotions in their interaction with the media. It examines how selective exposure to the media is linked to how political emotions influence this process. The research reveals an emotional fracture in media consumers through anger. It is also observed that positive emotions towards leaders and political parties are activated in like-minded media consumers, whereas negative emotions arise when interacting with oppositely oriented media. The importance of positive emotions such as hope and their contrast with negative aversive emotions is highlighted. This study shows how political emotions influence the interaction between citizens and the media. The findings highlight the relevance of emotions in the formation of political attitudes and the configuration of media preferences, contributing to the discussion on the relationship between the media and emotions in contemporary society.

**Keywords:** emotions; selective exposure; mass media



## 1. Introduction

In this work, we will explore the intersection between the media and the emotions activated in citizens. We will explore how media consumption defines how people experience and express emotions. In doing so, we aim to shed light on the complex relationship between media and emotions, as well as understand their influence on today's society and its implications for the future.

In recent years, Spain has experienced an increase in political polarization due to multiple factors. The economic crisis, the question of Catalonian independence, and party fragmentation are key elements. In addition, exacerbated political discourses and the influence of the media have intensified this polarization. The proliferation of social networks has accentuated the phenomenon by creating information bubbles, where people connect mainly with those who share their opinions, thus limiting exposure to different perspectives. All of this has contributed to dividing society into extreme positions, often driven by an emotional charge that reinforces political beliefs and attitudes.

The expansion and use of digital infrastructures in Spain have generated opportunities both for the expansion of traditional media and for the emergence of new actors in the media landscape. A clear example of this expansion is the implementation of digital terrestrial television (DTT), which has led to a notable increase in the number of television channels and information programs in the media. Likewise, radio has experienced a similar use of digital infrastructures, without relying exclusively on transmission over the airwaves to use new platforms such as the internet for its dissemination. In the case of newspapers, these infrastructures have provided new media, driving the transition from print to digital.

The Spanish transition has left a deep mark on the country's media landscape. The change in political regime is also manifested in aspects such as censorship and the issuance of new licenses. Especially notable and emblematic were the newspapers *Diario16* and *El País*, both founded in 1976 after having been limited during Franco's period. These

developments have a significant impact on the quality of democracy. In political theory, it has been argued that the media play a fundamental role as fundamental instruments for democracy. Habermas (1996) and Dahl (2012) argue that the media are the basis by which citizens interact with the state, presenting a range of issues and elevating the most relevant ones (McCombs and Shaw 1991; Althaus and Tewksbury 2002) for discussion in the public realm. In this way, freedom of expression and freedom of the press stand as essential elements of any democracy. However, the quality of democracy cannot be assessed solely by these parameters. Accessibility to quality information and cultural aspects of the nation are also important in measuring the quality of democracy. Interest in politics, political participation, and trust in democratic institutions are also important. However, the construction of solid public opinions could be conditioned by the nature of the media system. Some important characteristics of the media model for a democracy are the degree of plurality, the degree of competitiveness, and the atomization of the media in large media corporations.

In the age of digital information and constant interconnection, the media plays a fundamental role in the construction of collective perception and formation of opinions in contemporary society. This constant interaction has radically transformed the way people experience and process their environments, especially politics. The main risk of these new information channels is the proliferation of biased or false news or information. The need to be the first to publish information, which means passing fewer filters to verify information, as well as the intention to deceive determines the proliferation of fake news. Therefore, fake news is a threat to the integrity of information in a democratic society. Fake news weakens participation and alienates citizens from politics. In this context, the symbiotic relationship between the media and emotions emerges with greater force as a vital research field of growing interest.

The wide variety of media options available, from traditional platforms to innovative digital spaces, raises fundamental questions about how people select, interact, and process the emotionally charged information presented to them. As public opinion becomes increasingly shaped by media narratives, it is essential to understand how emotions intertwine with media selection, influencing individual and collective perception of reality. Therefore, one of the questions we ask ourselves is whether the choice of media and the way in which it is consumed end up significantly influencing how emotions are awakened and manifested in citizens.

In addition, the role of the media in political polarization and the creation of information bubbles or echo chambers depending on the environment that is analyzed has gained prominence in contemporary debates (Diaz Ruiz and Nilsson 2023; Jamieson and Cappella 2008; Nguyen 2020). Framing can lead to audience segmentation into groups with increasingly polarized political views. This polarization not only reflects a gap in the perception of events but also incites and fuels emotions such as fear and attitudes such as indignation, as well as affirmation, which can influence political decision-making and social dynamics (Lagares et al. 2022b).

## 2. Theoretical Framework

### 2.1. Media Consumption and Selective Exposure

Citizens preconsciously work on the coherence of their prejudices, avoiding all information that attacks their beliefs, values, or political affinities. The question is how the media influences the political behavior of citizens. The theory of limited media effects (Lazarsfeld et al. 1948; Klapper 1960) downplays them. The field of information psychology has extensively addressed individual biases in information consumption, in what has been termed "selective exposure" (Katz and Lazarsfeld 1970; Lazarsfeld et al. 1948).

Selective exposure explains that the media plays a catalytic role in the selection, attention, and retention of information. The media is no longer considered a. fundamental actor to manipulate or modify attitudes and behaviors; on the other hand, the effects it generates on citizens are minimal or even null. Under this new approach, audiences

unconsciously seek the reinforcement of their prejudices, the satisfaction or confirmation of their predispositions and opinions. A related study has found that homogeneous focus groups show a greater bias toward supporting information compared to contradictory information (Schulz-Hardt et al. 2000). Today, we know that citizens who are more involved in political campaigns are more likely to seek partisan information (Chaffee and McLeod 1973). However, there is also evidence that citizens increasingly select news, including non-political news, based on their partisan biases (Goldman and Mutz 2011; Iyengar and Hahn 2009; Stroud 2008). This monitoring or search for related information requires five necessary conditions: the variety of information sources that the individual can access, the quantity and quality of information available, the individual's interest in the subject of interest, and the intensity of the cognitive dissonance (Frey 1986).

Cognitive dissonance, as a mediator of the reinforcement of these predispositions, is one of the most influential psychological shortcuts. Cognitive dissonance is the internal tension that a person experiences when faced with the contradiction between their beliefs and values. This notion is based on the idea that people have an innate motivation to maintain consistency in their prejudices, behaviors, and beliefs. Consequently, it is a psychological construction that seeks to harmonize stimuli and responses. Thus, when the individual receives information, they try to fit it into its pre-established schemes; if the stimulus does not fit, a feeling of discomfort is generated (Festinger 1957).

However, we must consider some limitations of cognitive dissonance theory. The first weakness is its limitation to fully explain selective exposure, because without activating cognitive dissonance an individual can prefer related information. In other words, the choice of information could be carried out with the purpose of preventing possible cognitive conflicts in the future. Although cognitive dissonance is important, it is not a requirement for selective exposure to manifest (Stroud 2010). In addition, factors such as individual restrictions in information processing, the subjective perception of information quality, and the intrinsic need to search for answers, solutions, or truths in order to reach quick and desired conclusions, as well as the fear of making erroneous decisions (Kruglanski 1989), contribute significantly to the emergence and persistence of the phenomenon of selective exposure.

We should not ignore expressive behavior as a possible conditioning factor of selective exposure. Citizens tend to behave in politics in very different ways, even though their actions are the same. For example, a citizen may vote for a political party seeking to influence political decisions, but another citizen may do so as an expression of their values or identity (Hamlin and Jennings 2011). From this perspective, the idea of utility acquires a new meaning, expressive utility, which confirms the individual identity over a group. Thus, we can understand that expressive behavior is rational because it is self-serving (Hillman 2010). From this perspective, reading a specific newspaper, listening to a radio station, or watching the news on a given television channel could be associated with a form of identity or a symbolic reality of values and ideas that define the individual within a group. Expressive behavior is also analyzed at the level of social networks. Barnidge et al. (2018) have pointed out that there is a positive relationship between the new uses of social networks and this type of behavior, since they are more common among those who use online platforms.

Likewise, we consider it important to warn about the connections between selective exposure and media bias. The evidence indicates that some media outlets shape their editorial lines in favor of political parties. The use of vocabulary or the tone used by these outlets (Elejalde et al. 2018) define more favorable frameworks. This link between media bias and political parties has also been evident in the ideological spectrum of the extreme right. One of the particularities of these far-right media outlets is that they are not affected by perceptions of bias (Eberl 2019).

Furthermore, it has been shown how the media has shaped news to align with the prejudices of its audiences (Gentzkow et al. 2015), although this does not benefit all market participants (Gentzkow and Shapiro 2006). With this, the media has found a business

opportunity, detecting that political polarization frames increase the profitability of biased news (Bernhardt et al. 2008). On the contrary, competition between media outlets is essential in reducing media bias. Therefore, it is possible to intuit that the predisposition of citizens to consume news that reinforces their political prejudices and the inclination of certain media to adapt to the prejudices of their audiences leads to the construction of echo chambers.

*2.2. Media Consumption and Polarization*

The relationship between elites and the media may be one of the causes that contributes to a greater or lesser extent to the polarization of society. The media and elites struggle to introduce the issues and frameworks of news that reaches society. One of the characteristics of the Spanish media model is the high link to political power, for example, with political parties or governments. This relationship is characterized by economic exchange through advertising and institutional communication (De Miguel and Pozas 2009) by the granting of licenses in the audiovisual sector or by the degree of independence of journalists and the media. This model that relates political power to media power has been called "polarized pluralism" (Hallin and Mancini 2008).

The role played by political elites is central to our analysis, for example, in polarization (Fiorina et al. 2005; Jacobson 2003). This new problem raises other questions. Some researchers have devoted themselves to observing whether the polarization of elites has been transferred to the citizenry (see Jacobson 2003; Fiorina et al. 2005). However, the results are inconclusive. Other work has sought to find patterns through exposure to one type of media—television, radio, newspapers, the internet—and polarization (Druckman and Parkin 2005; Jones 2002; Bimber and Davis 2003).

However, there are numerous studies that show how selective exposure to the media plays in favor of greater polarization (Adams et al. 1985; Bimber and Davis 2003; Druckman and Parkin 2005; Jones 2002; Lavine et al. 2000; Mendelsohn and Nadeau 1996; Stroud 2007, 2010; Taber and Lodge 2006). Indeed, the connection between selective exposure, partisanship (Abramowitz 2015a, 2015b), and intensification of political attitudes has been validated by Mutz (2006). This is especially relevant if we link it to perceptions of an increase in the political bias of news among Spanish citizens (Farias and Roses 2009), which would clearly respond to the ideological and partisan alignments of the main Spanish publishers towards greater polarization (González and Novo 2012; Martínez Nicolás and Humanes 2012; van Dalen et al. 2012).

More polarized citizens are more likely to consume content close to their political positions (Levendusky 2013; Prior 2013; see Arceneaux et al. 2012), although this is not always the case. Stroud (2010) explains that, in certain situations, polarized citizens tend to consume information dissonant with their beliefs and prejudices because it further strengthens their opinions. This way of consuming information has its answer in the business model adopted by many publishers. More and more media outlets are accentuating their editorial lines, better segmenting their audiences, and skewing their information with two objectives: occupying a larger audience share and meeting the non-dissonant needs of audiences (Iyengar and Hahn 2009).

Most of these works are based on the idea of positional polarization (Kligler-Vilenchik et al. 2020; Yarchi et al. 2021), which consists of the assumption of extreme and incompatible preferences around a political issue (Abramowitz 2015a, 2015b; DiMaggio et al. 1996). This polarization is based on the attitudes of individuals towards political affairs and can be conceived in a static or dynamic way according to their change over time. However, it can also be interpretive polarization depending on framing (Baden 2015). On the other hand, we must consider the polarization based on emotions because of the rejection of the political adversary (McLaughlin et al. 2020). This idea of polarization should not be confused with affective polarization that is constructed with distance from another political faction. Affective polarization is based on partisan identification (Iyengar et al. 2012; Huddy and Yair 2021; Wagner 2021) and, far from reflecting an action of affections between partisans,

focuses on attitudes and therefore uses the fundamental basis of affectivity, which are the emotions that build such polarization (Lagares et al. 2022b).

*2.3. Emotions and Media Consumption*

The framing of the news has a clear effect on how it is perceived and can provoke a variety of emotional responses in people, from pride and fear to anxiety and anger. These emotions have a direct impact on how we process that information (Marcus et al. 2000).

The theory of affective intelligence (Marcus et al. 2000; Marcus 2002) and other studies that support it (Vasilopoulou and Wagner 2017; Vasilopoulos 2018; Vasilopoulos et al. 2019; Marcus et al. 2019) have established a model in which emotions, especially the negative ones related to anxiety, function as early warning signals. These emotions, such as anxiety, fear, worry, or anger, drive people to seek information with the aim of reducing uncertainty and making more informed decisions. Voters conditioned by these emotions feel the need to consume more information to reduce uncertainty about the political threat. However, the interpretation of these emotions has often led to an optimistic view of democracy, assuming that citizens would be better informed. The reality is that this interpretation does not consider the selection and quality of the information that citizens consume. In fact, low-quality or biased information can have negative consequences for democracy (Gadarian and Albertson 2014). On the other hand, emotions such as anger have also been linked to social division. Anger has been associated with fracture and establishes the foundations for political polarization and confrontation (Lagares et al. 2022b; Mo 2021).

Political polarization has been theorized in terms of confrontation in centrifugal contexts. Some studies suggest that aversive emotions, such as hatred, resentment, bitterness, or disgust, are key ingredients for this confrontation. Unlike anger, these aversive emotions lead to irreconcilable situations and represent the destructive part of politics. They are emotions that invite confrontation, fight, and elimination of the adversary. Some stages of aversive emotions include the expression of rejection, intolerance, and harm toward those who hold contrary opinions (Halperin et al. 2009; Bar-Tal and Teichman 2005; Halperin 2008; Demertzis 2006). Particularly, emotions such as hate have been defined as destructive emotions and have been linked to totalitarian political systems and contexts. Other emotions such as disgust have been defined as emotions linked to intolerance and rejection of those who are different and break social homogeneity. The main characteristic of these types of emotions is that they are irreconcilable. That is, for people who feel hatred, disgust, bitterness, or resentment towards something or someone, it is very difficult for them to feel a positive emotion. In other words, it is difficult for them to integrate or respect the object or person toward whom they feels those emotions (Mo 2021).

Another peculiarity is that unlike emotions focused on anxiety, aversive emotions do not develop in an environment of uncertainty. They form in political spaces that people recognize and find familiar. This means that its effect on political behavior reinforces the political ideas, values, and prejudices of each individual (Marcus et al. 2000).

Positively valence political emotions are also considered drivers of political predispositions. Pride, enthusiasm, hope, and tranquility are some of the emotions that have been studied in an empirical approach. In particular, pride and enthusiasm have been associated with political mobilization and the construction of partisan identification (Lagares et al. 2022a). From a constructivist perspective, hope is an emotion that is constructed by reference to the future (Lagares et al. 2022a). This means that to generate hope in citizens it is necessary to define a common and shared objective (Castells 2012). Furthermore, hope allows political action to be sustained over time. Thanks to these qualities, hope has been especially associated with new political parties and new leadership (Mo 2021). On the other hand, pride is an emotion that is built by reference to the past. This emotion is more common in the oldest followers of a political party. It is built in the past because it is necessary to reach a milestone or achieve success (Lagares et al. 2022a). These past successes are what keep pride alive, which is very useful to mobilize the political bases (Mo 2021). Enthusiasm is built in the present through exaltation. All in all, enthusiasm is the most important

emotion for political mobilization (Mo 2021). From a general perspective, positive emotions reinforce political positions, citizens' prejudices, and their political heuristics. This means that they are important emotions to understand the selection of media and to understand the consumption of ideologically proximate information.[1]

This work has a constructivist approach that understands political emotions as a process of social construction within a specific cultural context, which makes them changeable and fickle. One of the keys to this approach is that emotions are closely linked to a society's system of values and beliefs, that is, they condition what you feel and how you feel. On the other hand, the value system, prejudices, and attitudes are emotional characteristics, but at the same time, they are also constructed. In addition, cultural factors, for example, religion, are a direct expression of a society's emotions (Armon-Jones 1985). The constructivist approach helps us to better understand emotions and their role in political societies, as well as in information processing. That is why it is also important to interpret the concepts, semantics, and meanings attributed to emotions in their political context (Feldman 2018; Mead 1993).

For our study, constructivism provides an approach with which to approach the approach of citizens to the media. To that end, we must consider the citizen an active object in the generation of emotions. When a citizen accesses, selects, or is exposed to the media, they are organizing the process of generating emotions (Armon-Jones 1985; Averill 1980; Feldman 2018) that reinforce their predispositions.

This emotional configuration will be called the "emotional regime" when we refer to the set of emotions that are generated in each context. On the other hand, when we talk about how and how much those emotions are expressed in that context, we will refer to "emotional architecture" (Lagares et al. 2022b).

## 3. Analytical and Methodological Framework

More recent work on the implications of emotions in politics has reinforced theories that link the cognitive and emotional process in the decision-making process and its connection to contextual information. The main objective of this paper is to describe the emotional regimes of the media. As specific objectives, we will look for the emotions that influence selective exposure to the media. In addition, we will identify which emotions are more important, whether emotions towards political leaders or towards political parties.

Taking these objectives as a reference, we propose the following hypotheses, which we will try to contrast throughout the investigation:

**Hypothesis 1.** *The emotional regime of the media is defined by positive emotions towards leaders and political parties close to the editorial line and negative emotions towards leaders and parties that are far from the editorial line.*

**Hypothesis 2.** *Positive emotions towards leaders and political parties that are close to the editorial line and negative emotions towards leaders and political parties that are far from the editorial line explain the selective exposure.*

**Hypothesis 3.** *Emotions toward political parties are more important than emotions toward political leaders in media selection.*

To test our hypotheses, we designed a quantitative methodological approach in two phases: descriptive and inferential. In the first phase, we will carry out a descriptive analysis that will draw the emotional profile of consumers of newspapers, radio, and television. We will also seek to find out whether a greater consumption of information through newspapers, radio, and television is associated with greater emotional intensity. Finally, multivariate statistical techniques will be used to measure which emotions explain the selection of media to inform themselves about politics.

We will use the survey "Estudio Política y Emociones en España, Febrero 2021" (EPEE, February 2021) carried out by the Equipo de Investigaciones Políticas of the University of Santiago de Compostela. The survey has as its field of study the entire national territory. The survey design had a total of 1000 telephone interviews through the CATI system. These were conducted with Spanish citizens over 18 years of age and were distributed proportionally according to the criteria of sex and age. The margin of sampling error for the sample was +/− 3.1%.

The study provides a battery of 13 emotions (Appendix A) that we selected as independent variables. These emotions are grouped into three groups following the model of Marcus et al. (2000) (see Jaráiz et al. 2020). The first group is composed of positive emotions (pride, enthusiasm, peace of mind, and hope). The second group is composed of negative emotions linked to anxiety (anxiety, fear, worry, and anger). The third group is composed of very negative and therefore very extreme emotions (bitterness, hatred, resentment, contempt, and disgust). In the annexes, we show the list of variables used, their typology, and the level of measurement.

## 4. Results

Our analysis consists of three sections. First, we describe the regime and emotional architecture of the Spanish media model, paying special attention to the main emotions present in consumers of political information. Second, we look for associations between emotions and the frequency of consumption of political information. Finally, we try to explain which emotions determine the choice of media to inform oneself about politics.

### 4.1. Descriptive Analysis of the Emotional Regime of the Media

We begin the analysis by describing the regimes and emotional architectures of the Spanish media, paying special attention to the main publishers of newspapers, television channels, and radio stations (Tables 1–6). Overall, our initial analysis reveals two discernible emotional patterns among consumers of newspapers, television, and radio stations. The first one is related to the manifestation of emotions common to all leaders and parties, which allows us to identify a shared emotional regime. Among these emotions, anger and worry predominate. Both emotions belong to the dimension of anxiety and are activated by the detection of a political threat.

The main difference is that anger reflects the emotional fracture existing in Spanish politics, specifically towards leaders and political parties (Lagares et al. 2022a). The nature of anger in politics is the activation of radicalized behavior, for example, by confronting a threat (Carver and Harmon-Jones 2009; Lerner and Keltner 2000). Therefore, anger is *conditio sine qua non* of political fracture or polarization.

Concern is also a reflection of a state of mind that is limited to the general uncertainty that exists in citizens because of these same political actors (Mo 2021). These two emotions are widely present in each of the newspapers analyzed, and their architectures vary depending on the publisher.

The second pattern is associated with the ideological cleavage that determines the editorial line of each media outlet. Citizens who choose to be informed through newspapers with an editorial orientation close to the progressive left or right experience more favorable emotions towards leaders and political parties close to that faction, whereas they feel negative emotions towards leaders and parties of more conservative orientation. Similarly, those who choose publications with a conservative or liberal bent tend to experience positive emotions toward leaders and parties that share that perspective and negative emotions toward left-wing leaders and parties.

**Table 1.** Emotions towards the main Spanish political leaders based on newspapers.

| | El País | | | | | El Mundo | | | | | ABC | | | | |
|---|---|---|---|---|---|---|---|---|---|---|---|---|---|---|---|
| | Sánchez | Casado | Iglesias | Abascal | Arrimadas | Sánchez | Casado | Iglesias | Abascal | Arrimadas | Sánchez | Casado | Iglesias | Abascal | Arrimadas |
| Pride | 18.50% | 4.70% | 15.90% | 4.80% | 16.90% | 4.00% | 18.20% | 4.00% | 17.00% | 16.20% | 6.80% | 22.70% | | 25.60% | 23.80% |
| Fear | 9.90% | 14.00% | 23.80% | 51.00% | 8.10% | 21.00% | 7.10% | 48.00% | 23.00% | 6.10% | 34.10% | 6.80% | 45.50% | 11.60% | 7.10% |
| Hope | 35.10% | 10.00% | 23.80% | 6.80% | 24.30% | 8.00% | 39.40% | 4.00% | 24.00% | 34.30% | 13.60% | 45.50% | 4.50% | 34.90% | 40.50% |
| Anxiety | 12.60% | 16.70% | 15.20% | 24.50% | 8.10% | 20.00% | 9.10% | 18.00% | 8.00% | 4.00% | 25.00% | 11.40% | 25.00% | 11.60% | 9.50% |
| Enthusiasm | 19.20% | 3.30% | 10.60% | 4.80% | 13.50% | 3.00% | 21.20% | 3.00% | 13.00% | 17.20% | 4.50% | 25.00% | 2.30% | 18.60% | 26.20% |
| Anger | 37.10% | 41.30% | 36.40% | 54.40% | 23.00% | 55.00% | 32.30% | 53.00% | 31.00% | 23.20% | 68.20% | 29.50% | 63.60% | 16.30% | 21.40% |
| Hate | 6.60% | 5.30% | 4.60% | 16.30% | 4.10% | 12.00% | 4.00% | 15.00% | 5.00% | 2.00% | 15.90% | 6.80% | 13.60% | 2.30% | 4.80% |
| Contempt | 7.30% | 12.00% | 9.90% | 30.60% | 4.10% | 20.00% | 5.10% | 26.00% | 13.00% | 4.00% | 31.80% | 6.80% | 34.10% | 2.30% | 4.80% |
| Worry | 37.70% | 36.00% | 38.40% | 51.70% | 23.00% | 54.00% | 32.30% | 54.00% | 32.00% | 23.20% | 72.70% | 36.40% | 70.50% | 34.90% | 28.60% |
| Peace of mind | 29.10% | 8.00% | 13.90% | 6.10% | 16.90% | 3.00% | 27.30% | 2.00% | 14.00% | 22.20% | 11.40% | 34.10% | 2.30% | 18.60% | 23.80% |
| Resentment | 7.90% | 7.30% | 7.90% | 10.90% | 4.70% | 14.00% | 5.10% | 21.00% | 8.00% | 4.00% | 25.00% | 4.50% | 11.40% | | 2.40% |
| Bitterness | 7.30% | 8.70% | 6.60% | 16.30% | 5.40% | 16.00% | 5.10% | 19.00% | 7.00% | 3.00% | 27.30% | 13.60% | 22.70% | 7.00% | 4.80% |
| Disgust | 6.60% | 11.30% | 7.90% | 28.60% | 4.70% | 15.00% | 2.00% | 18.00% | 4.00% | 1.00% | 27.30% | 4.50% | 27.30% | 4.70% | 4.80% |

Source: authors' own creation based on the data of the "Estudio Política y Emociones en España, Febrero 2021," conducted by the political research team.

**Table 2.** Emotions towards the main Spanish political parties based on newspapers.

| | El País | | | | | El Mundo | | | | | ABC | | | | |
|---|---|---|---|---|---|---|---|---|---|---|---|---|---|---|---|
| | PSOE | PP | POD | VOX | C's | PSOE | PP | POD | VOX | C's | PSOE | PP | POD | VOX | C's |
| Pride | 30.50% | 6.60% | 16.60% | 4.00% | 15.90% | 6.00% | 22.00% | 1.00% | 19.00% | 14.00% | 11.40% | 27.30% | | 25.00% | 20.50% |
| Fear | 8.60% | 15.20% | 22.50% | 49.70% | 6.60% | 15.00% | 5.00% | 43.00% | 21.00% | 5.00% | 22.70% | 6.80% | 52.30% | 20.50% | 4.50% |
| Hope | 43.70% | 11.30% | 25.80% | 6.00% | 25.20% | 9.00% | 47.00% | 2.00% | 23.00% | 29.00% | 11.40% | 54.50% | 4.50% | 38.60% | 45.50% |
| Anxiety | 9.90% | 13.90% | 17.20% | 23.20% | 6.60% | 14.00% | 5.00% | 14.00% | 7.00% | 2.00% | 20.50% | 9.10% | 22.70% | 11.40% | 6.80% |
| Enthusiasm | 26.50% | 3.30% | 15.20% | 3.30% | 16.60% | | 23.00% | 2.00% | 12.00% | 14.00% | 6.80% | 25.00% | 2.30% | 15.90% | 22.70% |

**Table 2.** *Cont.*

| | *El País* | | | | | *El Mundo* | | | | | *ABC* | | | | |
|---|---|---|---|---|---|---|---|---|---|---|---|---|---|---|---|
| | **PSOE** | **PP** | **POD** | **VOX** | **C's** | **PSOE** | **PP** | **POD** | **VOX** | **C's** | **PSOE** | **PP** | **POD** | **VOX** | **C's** |
| Anger | 31.80% | 41.70% | 31.10% | 51.00% | 20.50% | 41.00% | 27.00% | 50.00% | 27.00% | 19.00% | 61.40% | 25.00% | 59.10% | 20.50% | 18.20% |
| Hate | 5.30% | 6.00% | 5.30% | 11.90% | 2.60% | 10.00% | 2.00% | 12.00% | 4.00% | | 11.40% | 2.30% | 13.60% | 2.30% | 4.50% |
| Contempt | 5.30% | 11.30% | 10.60% | 29.10% | 2.60% | 13.00% | 2.00% | 22.00% | 7.00% | 1.00% | 20.50% | 2.30% | 34.10% | 2.30% | 4.50% |
| Worry | 32.50% | 35.10% | 35.80% | 51.00% | 21.90% | 40.00% | 25.00% | 49.00% | 28.00% | 19.00% | 63.60% | 34.10% | 68.20% | 36.40% | 25.00% |
| Peace of mind | 34.40% | 8.60% | 18.50% | 4.00% | 15.20% | 4.00% | 32.00% | 2.00% | 15.00% | 20.00% | 13.60% | 38.60% | 2.30% | 20.50% | 25.00% |
| Resentment | 6.00% | 8.60% | 7.90% | 12.60% | 3.30% | 10.00% | 3.00% | 19.00% | 7.00% | 3.00% | 15.90% | 6.80% | 11.40% | | 2.30% |
| Bitterness | 5.30% | 7.30% | 4.60% | 14.60% | 4.00% | 14.00% | 4.00% | 18.00% | 6.00% | 1.00% | 18.20% | 6.80% | 22.70% | 4.50% | 4.50% |
| Disgust | 4.00% | 11.30% | 6.60% | 33.80% | 2.60% | 11.00% | 1.00% | 19.00% | 8.00% | | 18.20% | 4.50% | 27.30% | 4.50% | 4.50% |

Source: authors' own creation based on the data of the "Estudio Política y Emociones en España, Febrero 2021," conducted by the political research team.

**Table 3.** Emotions towards the main Spanish political leaders based on television.

| | La 1 | | | | | Antena 3 | | | | | Telecinco | | | | | LaSexta | | | | |
|---|---|---|---|---|---|---|---|---|---|---|---|---|---|---|---|---|---|---|---|---|
| | **Sánchez** | **Casado** | **Iglesias** | **Abascal** | **Arrimadas** | **Sánchez** | **Casado** | **Iglesias** | **Abascal** | **Arrimadas** | **Sánchez** | **Casado** | **Iglesias** | **Abascal** | **Arrimadas** | **Sánchez** | **Casado** | **Iglesias** | **Abascal** | **Arrimadas** |
| Pride | 18.80% | 7.60% | 10.80% | 5.80% | 13.60% | 9.40% | 13.70% | 4.30% | 15.00% | 18.10% | 20.20% | 13.80% | 8.50% | 9.90% | 11.00% | 18.50% | 3.80% | 14.60% | 0.80% | 13.80% |
| Fear | 4.70% | 8.50% | 21.10% | 34.60% | 3.40% | 23.50% | 8.60% | 43.80% | 27.50% | 6.60% | 10.60% | 7.40% | 26.60% | 28.60% | 4.90% | 8.50% | 11.50% | 16.90% | 41.90% | 7.30% |
| Hope | 37.60% | 20.90% | 15.00% | 9.10% | 21.40% | 16.20% | 36.90% | 8.20% | 22.30% | 32.70% | 24.50% | 13.80% | 9.60% | 9.90% | 15.90% | 38.50% | 6.20% | 30.80% | 3.10% | 20.30% |
| Anxiety | 7.50% | 9.50% | 10.80% | 16.80% | 3.90% | 22.60% | 11.20% | 26.20% | 12.40% | 5.30% | 12.80% | 8.50% | 12.80% | 13.20% | 7.30% | 10.00% | 13.10% | 10.00% | 17.10% | 7.30% |
| Enthusiasm | 11.70% | 8.10% | 7.50% | 4.30% | 12.60% | 5.10% | 17.60% | 4.30% | 13.30% | 15.90% | 10.60% | 5.30% | 6.40% | 4.40% | 3.70% | 12.30% | 4.60% | 13.10% | 0.80% | 11.40% |
| Anger | 42.70% | 38.90% | 44.10% | 45.70% | 27.20% | 55.10% | 29.60% | 53.20% | 31.30% | 18.60% | 41.50% | 33.00% | 45.70% | 39.60% | 20.70% | 33.80% | 47.70% | 28.50% | 45.70% | 30.90% |
| Hate | 3.30% | 3.80% | 4.20% | 9.60% | 1.00% | 11.50% | 3.00% | 15.90% | 7.70% | 1.80% | 5.30% | 6.40% | 6.40% | 12.10% | 8.50% | 6.90% | 6.20% | 4.60% | 14.00% | 3.30% |
| Contempt | 8.50% | 8.10% | 12.70% | 18.80% | 4.40% | 20.50% | 6.90% | 26.20% | 14.60% | 3.10% | 9.60% | 7.40% | 18.10% | 17.60% | 7.30% | 6.90% | 16.20% | 8.50% | 30.20% | 5.70% |
| Worry | 42.70% | 34.60% | 42.70% | 49.00% | 24.80% | 52.60% | 30.50% | 51.10% | 30.90% | 19.00% | 38.30% | 19.10% | 42.60% | 34.10% | 19.50% | 34.60% | 36.20% | 32.30% | 42.60% | 28.50% |
| Peace of mind | 23.90% | 11.80% | 9.90% | 4.80% | 14.60% | 9.00% | 23.20% | 3.90% | 12.40% | 19.50% | 22.30% | 16.00% | 8.50% | 5.50% | 12.20% | 30.80% | 7.70% | 14.60% | 3.10% | 12.20% |
| Resentment | 8.90% | 6.20% | 7.50% | 7.70% | 3.90% | 17.50% | 7.30% | 18.50% | 6.90% | 4.40% | 10.60% | 5.30% | 10.60% | 9.90% | 7.30% | 7.70% | 10.80% | 6.20% | 13.20% | 5.70% |
| Bitterness | 8.00% | 7.10% | 6.60% | 11.50% | 2.90% | 18.80% | 7.70% | 18.50% | 8.60% | 3.10% | 11.70% | 4.30% | 16.00% | 11.00% | 4.90% | 10.80% | 10.80% | 10.00% | 14.70% | 7.30% |
| Disgust | 6.60% | 5.70% | 7.50% | 12.50% | 2.40% | 13.70% | 4.30% | 18.90% | 11.20% | 2.70% | 8.50% | 7.40% | 10.60% | 20.90% | 3.70% | 6.90% | 13.80% | 6.20% | 31.00% | 5.70% |

Source: authors' own creation based on the data of the "Estudio Política y Emociones en España, Febrero 2021," conducted by the political research team.

**Table 4.** Emotions towards the main Spanish political parties based on television.

| | La 1 | | | | | Antena 3 | | | | | Telecinco | | | | | La Sexta | | | | |
|---|---|---|---|---|---|---|---|---|---|---|---|---|---|---|---|---|---|---|---|---|
| | PSOE | PP | POD | VOX | C's | PSOE | PP | POD | VOX | C's | PSOE | PP | POD | VOX | C's | PSOE | PP | POD | VOX | C's |
| Pride | 27.70% | 11.70% | 12.20% | 6.10% | | 13.70% | 19.20% | 3.80% | 14.10% | 15.00% | 24.50% | 17.00% | 7.40% | 7.40% | 9.60% | 29.20% | 3.80% | 13.80% | 0.80% | 9.20% |
| Fear | 3.80% | 8.50% | 21.10% | 31.50% | 4.00% | 18.80% | 8.10% | 39.70% | 26.50% | 5.10% | 8.50% | 9.60% | 27.70% | 27.70% | 4.30% | 5.40% | 13.80% | 13.10% | 40.80% | 5.40% |
| Hope | 43.20% | 26.30% | 16.40% | 9.40% | | 19.20% | 42.70% | 6.40% | 20.50% | 28.20% | 28.70% | 19.10% | 9.60% | 10.60% | 14.90% | 42.30% | 6.90% | 34.60% | 0.80% | 14.60% |
| Anxiety | 7.00% | 8.50% | 9.40% | 15.50% | | 17.50% | 9.00% | 23.90% | 11.50% | 5.10% | 8.50% | 5.30% | 10.60% | 6.40% | 3.20% | 6.20% | 12.30% | 8.50% | 16.20% | 5.40% |
| Enthusiasm | 15.50% | 9.90% | 10.30% | 4.20% | | 7.70% | 20.10% | 3.00% | 12.80% | 13.20% | 11.70% | 10.60% | 6.40% | 5.30% | 5.30% | 18.50% | 3.10% | 16.90% | 0.80% | 9.20% |
| Anger | 35.20% | 37.10% | 39.90% | 43.20% | 12.00% | 45.70% | 26.10% | 50.90% | 25.60% | 15.40% | 33.00% | 26.60% | 37.20% | 33.00% | 13.80% | 23.80% | 46.20% | 23.10% | 44.60% | 26.20% |
| Hate | 3.30% | 3.80% | 3.80% | 8.90% | | 10.30% | 2.60% | 14.50% | 6.00% | 1.70% | 4.30% | 5.30% | 4.30% | 6.40% | 4.30% | 3.10% | 6.20% | 3.80% | 11.50% | 2.30% |
| Contempt | 7.00% | 7.50% | 11.70% | 16.40% | | 16.20% | 4.30% | 22.20% | 10.70% | 3.40% | 4.30% | 3.20% | 12.80% | 14.90% | 3.20% | 4.60% | 13.10% | 5.40% | 28.50% | 3.80% |
| Worry | 32.90% | 34.30% | 39.40% | 45.10% | 4.00% | 42.70% | 25.60% | 48.30% | 29.90% | 15.80% | 29.80% | 18.10% | 35.10% | 30.90% | 12.80% | 24.60% | 31.50% | 26.90% | 38.50% | 23.10% |
| Peace of mind | 25.80% | 14.60% | 9.90% | 4.70% | | 10.30% | 26.50% | 3.00% | 13.70% | 15.80% | 20.20% | 17.00% | 6.40% | 5.30% | 11.70% | 37.70% | 7.70% | 19.20% | 3.10% | 7.70% |
| Resentment | 8.00% | 7.50% | 6.60% | 8.00% | 4.00% | 13.70% | 6.40% | 17.10% | 6.40% | 3.40% | 9.60% | 5.30% | 8.50% | 9.60% | 5.30% | 5.40% | 10.80% | 3.80% | 13.10% | 4.60% |
| Bitterness | 6.60% | 7.50% | 5.60% | 10.30% | | 16.20% | 5.60% | 17.10% | 6.00% | 3.00% | 9.60% | 1.10% | 12.80% | 9.60% | 1.10% | 7.70% | 10.00% | 6.20% | 13.10% | 6.90% |
| Disgust | 5.20% | 5.20% | 7.00% | 13.60% | | 9.80% | 4.30% | 18.80% | 9.80% | 3.00% | 5.30% | 5.30% | 8.50% | 16.00% | 2.10% | 4.60% | 16.90% | 5.40% | 38.50% | 4.60% |

Source: authors' own creation based on the data of the "Estudio Política y Emociones en España, Febrero 2021," conducted by the political research team.

**Table 5.** Emotions towards the main Spanish political leaders based on the radio.

| | La Ser | | | | | Onda Cero | | | | | COPE | | | | | RNE | | | | |
|---|---|---|---|---|---|---|---|---|---|---|---|---|---|---|---|---|---|---|---|---|
| | Sánchez | Casado | Iglesias | Abascal | Arrimadas | Sánchez | Casado | Iglesias | Abascal | Arrimadas | Sánchez | Casado | Iglesias | Abascal | Arrimadas | Sánchez | Casado | Iglesias | Abascal | Arrimadas |
| Pride | 22.90% | 4.30% | 17.90% | 2.20% | 14.20% | 10.70% | 7.10% | 4.80% | 9.60% | 19.30% | 2.60% | 19.80% | | 24.10% | 21.10% | 18.40% | 12.50% | 10.20% | 10.40% | 14.90% |
| Fear | 7.90% | 17.30% | 20.00% | 48.90% | 6.00% | 15.50% | 8.30% | 29.80% | 30.10% | 4.80% | 28.40% | 4.30% | 54.30% | 20.70% | 7.00% | 14.30% | 4.20% | 34.70% | 25.00% | 6.40% |
| Hope | 45.00% | 7.90% | 25.70% | 4.30% | 17.90% | 19.00% | 26.20% | 8.30% | 10.80% | 38.60% | 4.30% | 44.80% | | 31.90% | 31.60% | 36.70% | 14.60% | 16.30% | 12.50% | 23.40% |
| Anxiety | 9.30% | 15.80% | 15.00% | 23.70% | 7.50% | 19.00% | 13.10% | 19.00% | 22.90% | 6.00% | 25.00% | 7.80% | 25.00% | 7.80% | 4.40% | 8.20% | 10.40% | 10.20% | 10.40% | 6.40% |
| Enthusiasm | 17.10% | 5.00% | 14.30% | 3.60% | 9.70% | 7.10% | 9.50% | 2.40% | 6.00% | 15.70% | 0.90% | 23.30% | | 18.10% | 15.80% | 12.20% | 8.30% | 10.20% | 8.30% | 10.60% |
| Anger | 40.00% | 56.10% | 43.60% | 57.60% | 35.10% | 51.20% | 33.30% | 47.60% | 34.90% | 16.90% | 63.80% | 23.30% | 64.70% | 25.00% | 19.30% | 40.80% | 20.80% | 34.70% | 31.30% | 10.60% |
| Hate | 6.40% | 5.80% | 7.10% | 12.90% | 5.20% | 2.40% | 4.80% | 8.30% | 9.60% | 3.60% | 10.30% | 1.70% | 13.80% | 1.70% | 1.80% | 6.10% | 4.20% | 6.10% | 6.30% | 2.10% |

**Table 5.** *Cont.*

| | La Ser | | | | | Onda Cero | | | | | COPE | | | | | RNE | | | | |
|---|---|---|---|---|---|---|---|---|---|---|---|---|---|---|---|---|---|---|---|---|
| | Sánchez | Casado | Iglesias | Abascal | Arrimadas | Sánchez | Casado | Iglesias | Abascal | Arrimadas | Sánchez | Casado | Iglesias | Abascal | Arrimadas | Sánchez | Casado | Iglesias | Abascal | Arrimadas |
| Contempt | 7.90% | 13.70% | 8.60% | 26.60% | 9.00% | 14.30% | 10.70% | 23.80% | 22.90% | 7.20% | 23.30% | 4.30% | 31.00% | 6.90% | 4.40% | 12.20% | 10.40% | 16.30% | 10.40% | 6.40% |
| Worry | 32.10% | 36.70% | 32.10% | 50.40% | 24.60% | 56.00% | 38.10% | 50.00% | 44.60% | 30.10% | 69.00% | 30.20% | 63.80% | 31.00% | 23.70% | 55.10% | 29.20% | 42.90% | 31.30% | 19.10% |
| Peace of mind | 33.60% | 5.80% | 15.70% | 5.00% | 12.70% | 13.10% | 14.30% | 3.60% | 6.00% | 19.30% | 2.60% | 34.50% | | 16.40% | 20.20% | 22.40% | 18.80% | 8.20% | 8.30% | 21.30% |
| Resentment | 10.00% | 10.80% | 10.70% | 12.20% | 6.00% | 10.70% | 7.10% | 10.70% | 9.60% | 6.00% | 17.20% | 3.40% | 18.10% | 4.30% | 4.40% | 8.20% | 6.30% | 8.20% | 4.20% | 4.30% |
| Bitterness | 7.10% | 10.10% | 12.10% | 14.40% | 6.00% | 11.90% | 8.30% | 10.70% | 10.80% | 4.80% | 20.70% | 3.40% | 17.20% | 2.60% | 2.60% | 4.10% | 10.40% | 6.10% | 6.30% | 10.60% |
| Disgust | 5.00% | 10.10% | 6.40% | 28.80% | 6.00% | 7.10% | 6.00% | 10.70% | 13.30% | 3.60% | 20.70% | 1.70% | 23.30% | 2.60% | | 6.10% | | 10.20% | 6.30% | 6.40% |

Source: authors' own creation based on the data of the "Estudio Política y Emociones en España, Febrero 2021," conducted by the political research team.

**Table 6.** Emotions towards the main Spanish political parties based on the radio.

| | La Ser | | | | | Onda Cero | | | | | COPE | | | | | RNE | | | | |
|---|---|---|---|---|---|---|---|---|---|---|---|---|---|---|---|---|---|---|---|---|
| | PSOE | PP | POD | VOX | C's | PSOE | PP | POD | VOX | C's | PSOE | PP | POD | VOX | C's | PSOE | PP | POD | VOX | C's |
| Pride | 31.40% | 4.30% | 17.10% | 2.10% | 12.90% | 19.00% | 11.90% | 6.00% | 7.10% | 15.50% | 9.50% | 31.00% | | 25.90% | 18.10% | 26.50% | 10.20% | 10.20% | 10.20% | 18.40% |
| Fear | 6.40% | 19.30% | 17.90% | 45.70% | 5.70% | 9.50% | 9.50% | 28.60% | 29.80% | 4.80% | 20.70% | 4.30% | 50.00% | 19.80% | 6.00% | 12.20% | 6.10% | 24.50% | 20.40% | 4.10% |
| Hope | 49.30% | 8.60% | 27.90% | 2.90% | 19.30% | 26.20% | 31.00% | 10.70% | 8.30% | 32.10% | 8.60% | 56.90% | | 36.20% | 31.00% | 38.80% | 10.20% | 18.40% | 10.20% | 24.50% |
| Anxiety | 7.10% | 16.40% | 13.60% | 23.60% | 6.40% | 14.30% | 11.90% | 17.90% | 21.40% | 4.80% | 19.80% | 5.20% | 22.40% | 6.00% | 4.30% | 6.10% | 8.20% | 8.20% | 10.20% | 4.10% |
| Enthusiasm | 22.10% | 2.10% | 16.40% | 2.10% | 9.30% | 10.70% | 11.90% | 6.00% | 6.00% | 19.00% | 4.30% | 30.20% | | 17.20% | 12.90% | 18.40% | 6.10% | 14.30% | 6.10% | 12.20% |
| Anger | 32.90% | 55.00% | 36.40% | 56.40% | 29.30% | 41.70% | 33.30% | 40.50% | 31.00% | 15.50% | 48.30% | 18.10% | 57.80% | 23.30% | 13.80% | 40.80% | 24.50% | 40.80% | 24.50% | 14.30% |
| Hate | 2.90% | 6.40% | 4.30% | 10.70% | 2.10% | 3.60% | 4.80% | 2.40% | 7.10% | 1.20% | 9.50% | 1.70% | 12.90% | 2.60% | 1.70% | 4.10% | 2.00% | 6.10% | 6.10% | 2.00% |
| Contempt | 7.10% | 13.60% | 5.70% | 26.40% | 6.40% | 6.00% | 3.60% | 14.30% | 15.50% | 2.40% | 16.40% | 2.60% | 28.40% | 5.20% | 2.60% | 6.10% | 8.20% | 18.40% | 6.10% | 6.10% |
| Worry | 24.30% | 33.60% | 29.30% | 47.10% | 22.10% | 39.30% | 35.70% | 40.50% | 40.50% | 22.60% | 50.90% | 27.60% | 56.00% | 27.60% | 19.00% | 36.70% | 30.60% | 42.90% | 32.70% | 14.30% |
| Peace of mind | 40.00% | 5.00% | 19.30% | 3.60% | 10.00% | 11.90% | 17.90% | 4.80% | 4.80% | 17.90% | 6.00% | 45.70% | | 21.60% | 19.00% | 22.40% | 16.30% | 10.20% | 6.10% | 18.40% |
| Resentment | 7.10% | 9.30% | 9.30% | 13.60% | 2.90% | 7.10% | 7.10% | 9.50% | 8.30% | 4.80% | 12.90% | 3.40% | 17.20% | 2.60% | 2.60% | 2.00% | 6.10% | 6.10% | 6.10% | 4.10% |
| Bitterness | 7.10% | 10.00% | 9.30% | 14.30% | 5.00% | 8.30% | 4.80% | 10.70% | 9.50% | 2.40% | 15.50% | 3.40% | 17.20% | 2.60% | 1.70% | | 6.10% | 4.10% | 6.10% | 8.20% |
| Disgust | 4.30% | 9.30% | 5.70% | 32.10% | 2.90% | 2.40% | 4.80% | 9.50% | 10.70% | | 13.80% | 0.90% | 21.60% | 4.30% | 0.90% | | 6.10% | 8.20% | 8.20% | 4.10% |

Source: authors' own creation based on the data of the "Estudio Política y Emociones en España, Febrero 2021," conducted by the political research team.

Although anger and concern are two transversal emotions, emotional architectures change depending on this ideological fracture linked to the editorial line. They are more intense towards leaders and political parties contrary to the ideological and editorial line of the media. However, it is also noteworthy that the intensity continues to be important towards the leaders and parties that have a better accommodation to the editorial and ideological line of the media outlet. This means that anger and worry are activated in society. In addition, the fact that anger is also present towards leaders and parties among consumers of information a priori close to these political actors leads us to question the idea that this emotion acts as a mechanism of escape from politics and whether, on the contrary, it connects with politics and information, although from the extremes.

Another pattern that describes the emotional regime of the media—newspapers, television, and radio—is the activation of hope as the main positive emotion. Hope is closely related to the leaders and political parties that best connect their ideological position with the editorial line of the media. For example, the readers of *El País*, which is assumed to have a progressive editorial line and be close to the left, feel to a greater extent hope towards Pedro Sánchez (35.1%) and PSOE (43.7%). On the other hand, the readers of *El Mundo* or *ABC*, with an economically more liberal editorial line and from a more conservative social point of view, feel this emotion to a greater extent towards Pablo Casado, Santiago Abascal, and Inés Arrimadas, as well as towards their respective political parties.

We generally describe hope as an emotion built into the future (Lagares et al. 2022a; Mo 2021) and linked to new leadership (May 2021). Singularly, the high activation of this emotion towards political party leaders serves as a reinforcement of political predispositions and prejudices through the configuration of a desire or goal. That is, hope leads citizens to consume media (Jones et al. 2013), who expect their values and preconceived ideas to be satisfied and sustaining this behavior over time (Averill and Thomas-Knowles 1991).

Other positive-valence emotions activated in a remarkable way, in order of importance, are peace of mind and pride. Peace of mind is a pleasurable emotion (Cordaro et al. 2016) that is activated in the absence of threats. When an individual who feels calm towards a leader or political party consumes political information in a media outlet, they seek to reinforce that same mood and therefore do not want cognitive dissonance or contradiction with their prejudices. Pride works in a similar way. Citizens who feel pride in a leader or a party consume media to reinforce their political predispositions.

Following the reading of Lagares et al. (2022a) and Mo (2021), these citizens have built their identification with a leader or a political party in the past, and the information they consume in the present, which is not dissonant, continues to sustain that identification.

The emotional regime of the media—newspapers, television, and radio—is characterized by the activation of negative-valence emotions. One of the emotions most activated by leaders and parties among the media is fear. This emotion is activated to a greater extent towards leaders and parties that do not share the ideological line of the consumed media. Following the theory of affective intelligence (Marcus et al. 2000), these types of emotions relate to a greater consumption and processing of information (Vasilopoulos 2018) for the reduction of uncertainty generated by "political threats." One of the most notable examples is the presence of fear of Santiago Abascal and VOX among the readers of *El País*, the viewers of TVE or La Sexta, and among the listeners of La Ser, as well as towards Pablo Iglesias among the readers of *El Mundo* and *ABC*, the viewers of Antena 3, and the listeners of COPE.

The emotional geometry around negative-valence emotions is variable. As we have explained, there is an emotional regime shared by the presence of anger and concern; however, their architectures are different for each media outlet. Aversive emotions, the most extreme among negative emotions, lead us to consider different emotional fractures and the existence of emotional polarization.

From a transversal reading, we detect a group of aversive emotions that are predominant over the others—for example, contempt, bitterness, resentment, and disgust. One of the characteristics of aversive emotions is that they are not conciliatory, that is, they describe

a mood of absolute rejection that is not reversible (Mo 2021). As with the emotions of the anxiety dimension, the aversive emotions among citizens consume an editorial line that radically dissociates from leaders and parties. For example, readers of *El País*, viewers of La Sexta, and listeners of La Ser largely feel contempt and disgust towards Santiago Abascal and VOX. Another example is the contempt for Pedro Sánchez and Pablo Iglesias among the readers of *El Mundo* and *ABC*, the viewers of Antena 3, and the listeners of COPE.

The question that we cannot resolve here is whether polarization and fracture are built from the media or whether emotionally polarized citizens go to those media outlets to reinforce their moods and opinions.

However, the results seem to indicate that affective-based partisanship is linked to the selection of the most closely related medium. In general terms, emotions are activated to a greater extent towards political parties than towards their respective leaders considering each media outlet. These results are consistent with the conditioning of emotionally based partisan identification (Humanes 2014) (Lagares et al. 2022a; Mo 2021) with selective media exposure. Only in some exceptions are these emotions more accentuated towards their leaders. The reason we find is the identity base that positive emotions such as hope, pride, and enthusiasm have in the construction of party identification (Lagares et al. 2022a), especially for those more settled in the system, such as PSOE and PP. On the other hand, we are also aware that some emotions are more reactive and impersonal than others. Personal emotions are those directed toward someone with whom an individual identifies (Strawson 2008). For example, hyper-leadership such as that of Iglesias or Abascal makes certain ones such as fear or contempt be directed more towards their leaders than towards their respective political parties.

### *4.2. Tracking Information and Emotional Intensity*

In the previous pages, we defined selective exposure as the psychological orientation to the reinforcement of prejudices, values, and personal ideas. Tables 7 and 8 present an analysis of bivariate correlations between the frequency of monitoring information in newspapers, radio, and television with the emotional intensity towards leaders and political parties. We want to identify which emotions are most associated with increased media consumption. We also want to observe whether a higher frequency of monitoring political information through newspapers, radio, and television is associated with greater emotional intensity.

The results show an interesting initial pattern, with a greater number of significant correlations between the frequency of monitoring political information through newspapers and emotional intensities towards leaders and political parties. On the other hand, significant correlations between the frequency of monitoring information on radio and television with emotional intensities are minimal.

Regarding the degree of follow-up of political information through newspapers, the greatest number of correlations are concentrated with emotions towards Inés Arrimadas and Ciudadanos. These correlations, although weak, associate a greater consumption of political information in newspapers with a lower emotional intensity towards Arrimadas and Ciudadanos. These emotions are positive (enthusiasm and peace of mind) and negative in the dimension of anxiety and aversion.

The second group with the highest number of emotional correlations is Pablo Casado and PP. Similarly, we observe weak correlations towards positive emotions such as enthusiasm and negative emotions that involve both anxiety and aversiveness. It is necessary to point out that the strongest correlation is established with moderate strength between a greater follow-up of political information through newspapers and a lower intensity of hatred towards PP.

**Table 7.** Correlation analysis: degree of monitoring of political information and emotional intensities towards political leaders.

| | Emotions Sánchez | | | Emotions Casado | | | Emotions Iglesias | | | Emotions Abascal | | | Emotions Arrimadas | | |
|---|---|---|---|---|---|---|---|---|---|---|---|---|---|---|---|
| | Newspapers | Radio | TV | Newspapers | Radio | TV | Newspapers | Radio | TV | Newspapers | Radio | TV | Newspapers | Radio | TV |
| Pride | | | | | | | | | | | | | | | |
| Fear | | | | −0.215 * | | | | | | | | | | | |
| Hope | | | | | | | −0.194 * | | | | | | −0.172 * | | |
| Anxiety | | | | | | | −0.222 ** | | | | | | −0.280 * | | |
| Enthusiasm | −0.400 ** | | | −0.215 * | | | | | | | | | −0.330 ** | | |
| Anger | | | | | −0.140 ** | | −0.137 ** | | | | | | −0.133 * | −0.135 * | |
| Hate | | | | | | | −0.220 * | | | −0.212 * | | | −0.482 ** | | |
| Contempt | | | | −0.267 ** | | | | | | −0.144 * | | | −0.293 * | | |
| Worry | | | −0.101 * | −0.121 * | | | −0.135 ** | | | | | | −0.149 * | −0.150 * | |
| Peace of mind | −0.148 * | | | | | | −0.210 * | | | | | | −0.266 ** | | |
| Resentment | | | | | | | | | | −0.259 * | | | −0.320 * | | |
| Bitterness | | | | | | | | | | | | | −0.396 ** | | |
| Disgust | −0.263 * | −0.209 * | | −0.327 ** | | | | | | −0.169 * | | | −0.330 * | | |

Source: authors' own creation based on the data of the "Estudio Política y Emociones en España, Febrero 2021," conducted by the political research team. ** $p < 0.01$; * $p < 0.0$.

**Table 8.** Correlation analysis: degree of monitoring of political information and emotional intensities towards political parties.

| | Emotions PSOE | | | Emotions PP | | | Emotions Podemos | | | Emotions VOX | | | Emotions C's | | |
|---|---|---|---|---|---|---|---|---|---|---|---|---|---|---|---|
| | Newspapers | Radio | TV | Newspapers | Radio | TV | Newspapers | Radio | TV | Newspapers | Radio | TV | Newspapers | Radio | TV |
| Pride | −0.142 * | | | | | | | | | | 0.286 * | | | | |
| Fear | | | | −0.244 ** | | | | | | | | | −0.292 * | | |
| Hope | −0.126 * | | | | 0.160 * | | | | | | | | | | |
| Anxiety | | | | | | | −0.254 ** | | | | | | −0.343 * | | |
| Enthusiasm | −0.255 ** | | | | −0.239 * | | | −0.240 * | | | | | −0.252 * | | |
| Anger | | | | | −0.139 * | | −0.138 ** | | | | −0.115 * | | −0.147 * | −0.170 * | |
| Hate | | | | | −0.402 ** | | | | | −0.231 * | | | −0.364 * | | |
| Contempt | | | | | −0.302 ** | | | | | | | | −0.337 * | | |
| Worry | | | | | −0.149 * | | −0.109 * | | | | | | −0.243 ** | −0.214 ** | |
| Peace of mind | −0.199 ** | | | | | | −0.231 * | −0.264 ** | | | | | | | |
| Resentment | | | | | | | | | | −0.232 * | | | | | |
| Bitterness | | | | | 0.241 * | | | | | | | | −0.359 * | | |
| Disgust | | −0.289 * | −0.248 * | 0.300 ** | | | | | | | −0.206 ** | | | | |

Source: authors' own creation based on the data of the "Estudio Política y Emociones en España, Febrero 2021" conducted by the political research team. ** *p* < 0.01; * *p* < 0.05.

The next group with the highest number of correlations is Pablo Iglesias and Podemos. Among the correlations, we find associations with positive emotions such as hope towards Pablo Iglesias and peace of mind towards Podemos. Among the negative emotions, we find negative emotions of the anxiety dimension as well as the aversion dimension. All these correlations are weak.

The greatest number of correlations with positive emotions are towards the leadership of Pedro Sánchez and PSOE—for example, enthusiasm and peace of mind towards Pedro Sánchez or pride, hope, enthusiasm, and peace of mind towards PSOE. The strongest correlation is between the follow-up of information through the newspaper and enthusiasm for Pedro Sánchez (0.400). The only negative emotion is disgust towards Pedro Sánchez, indicating a negative correlation between a greater consumption of political information in the newspapers and an extreme rejection of the socialist leader.

The leader and party that present significant correlations only with negative emotions belong to Abascal and VOX, respectively. For example, a greater consumption of political information through newspapers is linked to lower intensities of anger, contempt, resentment, and disgust towards Abascal and a lower intensity of hatred and resentment towards VOX.

On the other hand, a greater degree of monitoring of political information through the radio is associated with a lower intensity of disgust towards Pedro Sánchez and towards PSOE. This high degree of follow-up through the radio is related to a lower intensity of anger towards Casado but with a lower intensity of hope towards PP. However, there are no correlations between the monitoring of political information through the radio and emotions towards Pablo Iglesias but rather towards Podemos. This correlation is positive and establishes a greater follow-up of political information and a lower intensity of enthusiasm and peace of mind towards Podemos. There are also no correlations with emotions towards Abascal but rather towards VOX, specifically with a lower intensity of pride, anger, and disgust. Finally, a greater follow-up of political information through the radio is associated with a lower intensity of anger and concern towards Arrimadas and Ciudadanos.

In the analysis of television, we only find significant correlations between the monitoring of political information and emotions towards Pedro Sánchez and PSOE. These emotions are negative valence. For example, a greater follow-up of political information through this medium is positively associated with a lower intensity of concern towards Pedro Sánchez and a greater intensity of disgust towards PSOE.

With the above analysis, we can consider that a greater follow-up of political information is associated with a lower affective intensity. However, the results show something else. The number of significant associations between the degree of follow-up to political information in newspapers and emotions suggests that news readers are more informed citizens and less susceptible to developing high emotional intensities. These results also inform the importance that newspapers still have today in political campaigns, because when you campaign in newspapers you reach a very specific audience.

The results also report on the characteristics of the profile of radio and television consumers. Tables 7 and 8 show that there are almost no significant relationships between the monitoring of information on television and emotions. This means that television is the media outlet that allows you to reach more people, but it is also a more heterogeneous audience. We also observe this in two specific media, Telecinco and Onda Cero.

### 4.3. Media Selection and Emotions: An Inferential Analysis

After having described the emotional presence towards leaders and political parties according to the media, we also want to verify the association between the frequency of information consumption and its association with emotional intensity. Next, we show which emotions explain the selection of the media to be informed. We decided to apply the binary logistic regression technique with the logit link function to determine which emotions influence selection. For the modeling, we decided to incorporate only emotional

variables towards the leaders and parties. We made this decision for two reasons. The first is that affective variables incorporate the partisan dimension and the leadership dimension. The second is that the global adjustment value of the model ($R^2$) indicates the degree of total variance explained by this type of variable.

The data shown in Table 9 indicate that emotions play an important role in media selection. The selection of the newspaper *El País* for information on politics is explained by 19.8% (pseudo $R^2$: 0.198) of the effect of political emotions towards leaders and political parties. However, it is not only emotions towards leaders and political parties that influence this selection. The ideological position of these leaders and parties also draws the gap that determines the selection of the newspaper and the other media, which we analyze in the following pages.

**Table 9.** Binary logistic regression model for the emotional explanation of newspaper selection.[2]

| | *El País* | *El Mundo* | *ABC* |
|---|---|---|---|
| Peace of mind Sánchez | | −2.011 ** (0.739) | |
| Disgust Sánchez | | | 1.917 *** (0.521) |
| Resentment Iglesias | | | −1.933 ** (0.667) |
| Anger Abascal | | | −1.146 * (0.459) |
| Enthusiasm PSOE | 0.813 ** (0.275) | | |
| Hope PSOE | | −0.891 * (0.381) | −1.226 * (0.535) |
| Worry PSOE | | | 0.862 * (0.382) |
| Enthusiasm PP | −1.281 * (0.568) | | |
| Hope PP | −1.008 ** (0.341) | 0.800 ** (0.250) | 0.769 * (0.374) |
| Hope Podemos | | −1.572 * (0.741) | |
| Anxiety Podemos | 0.829 ** (0.335) | −1.009 * (0.407) | |
| Bitterness Podemos | −1.513 ** (0.524) | | |
| Resentment Podemos | | 1.142 ** (0.397) | |
| Fear VOX | 0.532 * (0.218) | | |
| Disgust VOX | 0.665 ** (0.238) | | |
| Enthusiasm C's | 0.894 ** (0.340) | | |
| Hope C's | | | 0.923 * (0.381) |
| Contempt C's | −1.826 ** (0.575) | | −3.254 *** (0.317) |
| (Constant) | −1.619 *** (0.150) | −1.631 *** (0.162) | |
| $R^2$ Nagelkerke | 0.198 (19.8%) | 0.201 (20.1%) | 0.241 (24.1%) |

Source: authors' own creation based on the data of the "Estudio Política y Emociones en España, Febrero 2021," conducted by the political research team. *** $p < 0.001$; ** $p < 0.01$; * $p < 0.05$.

The choice of *El País* compared to the other newspapers is the most heterogeneous from an ideological perspective. Enthusiasm is the emotion that best explains the selection of this newspaper. The emotion that best explains the selection of *El País* as the main newspaper for information on politics is the enthusiasm for Ciudadanos (2.444) and PSOE (2.254). These results are consistent with those analyzed in Tables 1–6 and reinforce the idea that media selection is consistent between journalistic opinions and approaches (Humanes 2014).

On the other hand, the emotions that reinforce to a greater extent the selection of *El País* are of negative valence, with two linked to anxiety and one to aversion. Reading *El País* is an individual behavior influenced by anxiety towards Podemos (2.291) and fear of VOX (1.702) but also by the rejection generated by VOX through disgust (1.945).

The other emotions that build the decision to read *El País* are presented with a negative sign, that is, because those emotions towards leaders and parties are not activated. In addition, we must emphasize that the weight of these emotions is less. For example, not feeling hope (0.365) and enthusiasm (0.2789 towards PP increases the probability of reading *El País*. It is also interesting to observe the affective complexity that structures the decision, because the fact of feeling an emotion of anxiety does not mean that aversion works in the same way. This is the case of the bitterness towards Podemos, whose absence of emotion positively explains (0.220) the choice of *El País*. This same effect is also present in the model with the effect of contempt (0.161) towards Ciudadanos.

The choice of the newspaper *El Mundo* to be informed about politics is explained by 20.1% (Pseudo R2: 0.201) from the affective variables present in the model. The variable with the greatest explanatory power is a negative emotion. In fact, the probability of reading *El Mundo* increases by 3.133 points if a citizen feels resentment towards Podemos. It is this sense of threat towards the populist left party that determines to a greater extent the selection of the media outlet and that is consistent with other emotional readings of information processing (Vasilopoulos 2018; Marcus et al. 2000).

However, not only negative emotions determine this behavior but also positive emotions. In the same way that we observe with *El País*, the hope towards PP (2.226) explains the selection of *El Mundo* to informs oneself about politics. The presence of hope is also revealing because it is an indication of the profile of the reader of *El Mundo* compared to the reader of *El País*. Hope explains two things. First, citizens who have joined the party more recently are those who consume this information in order to strengthen their expectations, which is consistent with selective exposure theory (Humanes 2014). Secondly, the fact that pride or enthusiasm towards PP is not significant reveals that there is no party mobilization that seeks related political information to reinforce its political predispositions.

Finally, Table 9 shows how not all variables have a positive effect, and, in addition, the weight of these variables is significantly lower in the explanation of the model. However, we can extract some readings. The first is that the selection of *EL Mundo* as an informative newspaper is also explained by the absence of positive emotions towards Pedro Sánchez and PSOE. For example, the absence of peace of mind towards the socialist leader (0.134) and the absence of hope towards the PSOE leader (0.410), explain the dependent variable positively. On the other hand, the absence of anxiety towards Podemos (0.365) explains the choice of *El Mundo* to inform oneself about politics. To put it another way, reading *El Mundo* is not explained in terms of identifying a threat in Podemos but in terms of rejection.

The selection of the newspaper *ABC* to inform oneself about politics represents a particular and differentiated emotional regime. First, the model is explained by 24.1% (pseudo R2: 0.241) of the set of independent variables that are significant. As with the *El Mundo* model, the variable that most explains the choice of *ABC* as an informative newspaper is an aversive variable. In this case, feeling disgust towards Pedro Sánchez (6.799) increases the probability of reading *ABC* versus not feeling this emotion. If reading *El Mundo* is explained by the rejection of Podemos, reading the newspaper *ABC* is explained by the rejection of Pedro Sánchez. This is important because it is the first model in which an emotion towards a political leader is significant, which explains the degree of polarization and rejection existing among *ABC* readers towards the socialist leader. These results

warn us of how the most extreme emotions also work in reinforcing prejudices towards political adversaries.

The second most explanatory variable is the hope towards Ciudadanos (2.518), increasing the probability of reading that newspaper. This same effect is present in the hope towards PP (2.158). The presence of hope towards PP and Ciudadanos leads us to propose the same reading that we did among the readers of *El Mundo* and the absence of emotions such as pride and enthusiasm. The third most important variable is concern for PSOE. If Pedro Sánchez generates rejection among *ABC* voters, the Socialist Party represents the threat. Feeling concerned towards PSOE (2.367) increases the probability of reading *ABC* to learn about politics.

It is important to highlight the significance of positive emotions towards Abascal and VOX in the models of *El Mundo* and *ABC*. However, choosing *ABC* is also explained by the absence of fear towards the far-right leader (0.115). The choice of *ABC* also explains whether Abascal's leadership is a threat. On the other hand, the absence of hope towards PSOE (0.293) and resentment towards Iglesias (0.145) also determine the selection of *ABC* to inform oneself about politics.

Table 10 shows the results for the emotional explanation of the selection of television to inform oneself about politics. From a general perspective, the models sustain a greater emotional complexity in the choice of a television channel to inform oneself about politics. This complexity is measured by the number of significant emotions.

The choice of Antena 3 is explained by 23.6% (pseudo R2: 0.236) of significant emotions. The most important emotion for the explanation of the model is the fear towards Pedro Sánchez (2.570), which expresses the identification of a threat in the leader of PSOE and, therefore, increases the probability of consuming political information in Antena 3. This figure is consistent with the absence of peace of mind towards Pedro Sánchez (0.393). On the other hand, the most important positive emotion is the hope towards PP (2.119). Again, the absence of positive emotions such as enthusiasm or pride is also explanatory of the emotional regime of the consumers of this television channel.

Other negative emotions that explain being informed by Antena 3 are anxiety towards Podemos (1.845) and hatred towards Pablo Iglesias (1.815). As with newspapers such as *El Mundo* or *ABC*, identifying Podemos as a threat increases the likelihood of consumption of this media outlet. Similarly, rejection of Pablo Iglesias through hatred (1.815) increases the likelihood of watching Antena 3. To close the model, we must consider the non-activation of emotions such as concern for Inés Arrimadas (0.576), anger towards VOX (0.568), hope towards PSOE, or contempt for PP (0.287).

The Telecinco model is explained by 12.0% (pseudo $R^2$: 0.120) of the significant variables after the debugging process. The emotional regime of Telecinco's viewer is very complex and presents emotions that are apparently contradictory. However, the results are a demonstration of the heterogeneity of the consumer who watches this television channel to stay informed. The most important variable is hatred towards Pablo Casado (5.673). This is a sign of aversion, the rejection that some spectators feels towards the leader of PP, but that does not mean that they feel concern or see a threat in Casado (0.438). However, feeling pride towards Pablo Casado (3.714) also increases the likelihood of watching Telecinco, which represents the other part of the audience.

The presence of bitterness towards Pablo Iglesias represents the third most important variable. The fact of feeling bitterness towards Iglesias (2.678) increases the probability of watching Telecinco for information. The other most important positive emotion is peace of mind towards Ciudadanos. This means that feeling calm towards Ciudadanos increases the probability of selecting Telecinco as the main television network to inform oneself about politics.

The absence or non-activation of positive emotions towards right-wing leaders and parties also increases the probability of watching Telecinco, for example, when one feels enthusiasm towards Arrimadas or hope towards PP. In addition, bitterness towards PP (0.083) has a negative effect on the consumption of this television channel.

**Table 10.** Binary logistic regression model for the emotional explanation of television selection.

| | Antena 3 | Telecinco | La Sexta | TVE |
|---|---|---|---|---|
| Peace of mind Sánchez | −0.934 ** (0.305) | | | |
| Anxiety Sánchez | | | | −0.804 * (0.327) |
| Fear Sánchez | 0.944 ** (0.273) | | | −1.087 ** (0.397) |
| Pride Casado | | 1.312 ** (0.458) | | |
| Worry Casado | | −0.825 ** (0.316) | | |
| Hate Casado | | 1.736 ** (0.579) | | |
| Hate Iglesias | 0.596 + (0.336) | | | |
| Bitterness Iglesias | | 0.985 * (0.391) | | |
| Worry Abascal | | | | 0.757 *** (0.181) |
| Disgust Abascal | | | | −0.688 ** (0.260) |
| Enthusiasm Arrimadas | | −2.460 ** (0.774) | | 0.560 * (0.283) |
| Worry Arrimadas | −0.552 * (0.229) | | | |
| Hope PSOE | −0.573 * (0.225) | | | 0.500 ** (0.179) |
| Hope PP | 0.751 *** (0.196) | −1.139 ** (0.392) | −1.144 ** (0.367) | |
| Contempt PP | −1.249 ** (0.485) | | | |
| Bitterness PP | | −2.491 * (1.083) | | |
| Anxiety Podemos | 0.612 * (0.270) | | | |
| Hope Podemos | | | 0.851 *** (0.233) | |
| Hope VOX | | | −2.138 * (1.023) | |
| Anger VOX | −0.566 ** (0.202) | | | |
| Disgust VOX | | | 0.923 *** (0.226) | |
| Peace of mind C's | | 0.915 * (0.431) | | |
| Fear C's | | | | −1.338 * (0.553) |
| Contempt C's | | | −1.176 * (0.513) | |
| (Constant) | −0.837 *** (0.130) | −1.962 *** (0.152) | −1.883 *** (0.139) | −1.332 *** (0.131) |
| $R^2$ Nagelkerke | 0.236 (23.6%) | 0.120 (12.0%) | 0.160 (16.0%) | 0.123 (12.3%) |

Source: authors' own creation based on the data of the "Estudio Política y Emociones en España, Febrero 2021," conducted by the political research team. *** $p < 0.001$; ** $p < 0.01$; * $p < 0.05$; + $< 0.10$.

The choice of La Sexta as the preferred television channel to watch the news is explained by 16.0% (Pseudo $R^2$: 0.160) of significant emotions. The variable with the most explanatory power is a negative emotion, disgust towards VOX. Feeling this emotion about the far-right party increases the probability (2.518) of choosing La Sexta for information.

The second most important variable of the model is hope towards Podemos. This is the only positive emotion with a positive effect on the dependent variable. Feeling hope towards Podemos (2.343) increases the probability of using La Sexta to stay informed about politics. These results are also consistent with the non-activation of hope towards VOX (0.118) or towards PP (0.318), whose effect is negative on the dependent variable. We close the explanation of the model with the negative effect of bitterness towards Ciudadanos on the choice of La Sexta as the main media outlet.

The last TV channel we analyzed is Televisión Española (TVE). The selection of public television is explained by 17.4% (pseudo $R^2$: 0.174) of the significant variables in the model. The variable that best explains the model is concern towards Santiago Abascal (2.131). Identifying the leader of VOX as a threat, which builds a mood of concern, increases the likelihood of consuming this TV channel. As positive emotions, we observe enthusiasm towards Inés Arrimadas (1.750). Feeling enthusiastic towards the leader of Ciudadanos increases the probability of selecting TVE to learn about politics. On the other hand, feeling hope towards PSOE (1.649) generates the same effect.

The inactive emotions that explain the selection of TVE for information about politics are fear (0.337) and anxiety (0.448) towards Pedro Sánchez. The two emotions belong to the dimension of anxiety and are therefore activated in conditions of uncertainty or threat. The negative sign explains that the absence or non-activation of these emotions towards the socialist leader is a necessary condition for the choice of TVE to inform oneself about politics. The same happens with the fear of Ciudadanos (0.264). Finally, the absence of emotional rejection towards VOX, in this case, through disgust, explains watching TVE to learn about politics.

Table 11 presents the results for the emotional explanation of the selection of a radio station to inform about politics. The selection of Cadena SER to be informed about politics is explained by 17.4% (pseudo $R^2$: 0.174) of the significant emotions of the model. The three most important emotions are negative. The most important variable is disgust towards VOX, which shows that the rejection of the far-right party explains in a positive way the selection of Cadena Ser to stay informed. On the other hand, anxiety (2.233) and anger (2.233) towards Partido Popular have the same effect on the dependent variable. The most important positive emotions are hope for Pedro Sánchez (1866) and Peace of mind for Podemos (1.851).

However, it is not only the rejection of VOX that explains listening to Cadena SER but also the feeling of fear towards Santiago Abascal (1.792). The emotions that have a negative effect on the dependent variable are hatred towards Abascal (0.303), disgust towards PP (0.383), concern towards PSOE (0.443), and hope towards PP (0.380).

The selection of Cadena COPE to stay informed about politics is explained by 29.7% (pseudo R2: 0.297) of the emotions that are significant after debugging the model. The most important emotion of the model is positive valence. Feeling hope towards PP (4020) and towards VOX (2.784) explains in a positive way the selection of COPE Chain to inform oneself about politics. Contrary to hope, concern for Sánchez (2331), as well as the absence of peace of mind towards PSOE (0.290) or hope towards the socialist leader (0.202), explain why citizens choose COPE as the preferred station to learn about politics. Likewise, not feeling disgusted towards VOX (0.096) explains in a positive way listening to COPE.

The choice of Onda Cero for the monitoring of political information is explained by 11.7% (pseudo $R^2$: 0.117). Anxiety towards VOX (4.672) is the variable with the greatest explanatory power. Identifying VOX as a political threat helps citizens choose Onda Cero for information. The second most important variable is contempt for Pablo Iglesias (2.527).

The first positive emotion is hope for Inés Arrimadas. (1.844). This emotion builds the decision to listen to Onda Cero. Finally, not feeling disgust towards VOX (0.432), feeling

peace of mind towards PSOE (0.358), and feeling hatred towards Podemos (0.112) explain the decision to listen to Onda Cero to learn about politics.

**Table 11.** Binary logistic regression model for the emotional explanation of radio station selection.

|  | Cadena SER | COPE | Onda Cero | RNE |
|---|---|---|---|---|
| Hope Sánchez | 0.624 ** (0.213) | −1.599 ** (0.489) |  |  |
| Worry Sánchez |  | 0.846 *** (0.239) |  | 1.024 ** (0.329) |
| Bitterness Sánchez |  |  |  | −2.592 ** (0.878) |
| Peace of mind Casado |  |  |  | 1.752 ** (0.534) |
| Anger Casado |  |  |  | −1.402 ** (0.455) |
| Bitterness Casado |  |  |  | 1.895 ** (0.665) |
| Contempt Iglesias |  |  | 0.927 ** (0.312) |  |
| Fear Abascal | 0.583 * (0.225) |  |  |  |
| Hate Abascal | −1.195 ** (0.369) |  |  |  |
| Hope Arrimadas |  |  | 0.612 * (0.268) |  |
| Anger Arrimadas |  |  | −0.689 * (0.332) |  |
| Peace of mind PSOE |  | −1.239 ** (0.477) | −1.027 * (0.399) |  |
| Worry PSOE | −0.814 ** (0.256) |  |  |  |
| Hope PP | −0.967 ** (0.330) |  |  | −2.781 *** (0.659) |
| Peace of mind PP |  | 1.391 *** (0.247) |  |  |
| Anxiety PP | 0.803 * (0.346) |  |  |  |
| Anger PP | 0.765 ** (0.225) |  |  |  |
| Disgust PP | −0.961 * (0.388) | −2.345 * (1.056) |  |  |
| Peace of mind Podemos | 0.616 * (0.294) |  |  |  |
| Anger Podemos |  |  |  | 0.896 * (0.358) |
| Hate Podemos |  |  | −2.186 ** (0.783) |  |
| Hope VOX |  | 1.024 *** (0.265) |  |  |
| Anxiety VOX |  |  | 1.542 *** (0.365) |  |
| Disgust VOX | 0.679 * (0.275) |  | −0.839 * (0.426) |  |
| Contempt VOX |  |  |  | −1.483 * (0.677) |
| (Constant) | −2.309 *** (0.176) | −2.578 *** (0.194) | −2.406 *** (0.172) | −3.062 *** (0.259) |
| $R^2$ Nagelkerke | 0.174 (17.4%) | 0.297 (29.7%) | 0.117 (11.7%) | 0.174 (17.4%) |

Source: authors' own creation based on the data of the "Estudio Política y Emociones en España, Febrero 2021," conducted by the political research team. *** $p < 0.001$; ** $p < 0.01$; * $p < 0.05$.

The model of Radio Nacional de España is explained by 17.4% (pseudo $R^2$: 0.174). The heterogeneity indicates the presence of the two most important emotions towards the same

leader, Pablo Casado, but of a different nature. The emotion with the greatest explanatory power is bitterness towards Casado (6.649). Citizens who feel this emotion are more likely to select RNE. The second most important emotion is peace of mind towards the leader of PP (5.765). Not only those who feel a rejection towards the PP candidate decide to listen to RNE but also those who feel that Casado does not represent a danger. This emotion represents a difference with respect to concern towards Pedro Sánchez (2.785), a feeling that increases the probability of listening to RNE.

Another negative emotion with significant weight is anger towards Podemos. Anger is one of the most important political emotions because it represents a fracture with the political class and, in this case, within the populist radical left party. With this, feeling anger towards Podemos (2.450) positively explains the decision to listen to RNE to learn about politics.

To finish the explanation of the model, the absence or non-activation of emotions such as bitterness towards Pedro Sánchez (0.075), anger towards Pablo Casado (0.246), and contempt towards VOX (0.227) have a negative effect on the dependent variable.

## 5. Conclusions

The results of the research presented represent a relevant contribution to studies on emotion and media. Works that have addressed theories of selective media exposure have explained how people primarily seek and consume information that supports their political views. In fact, some of these works have considered partisan identification as the fundamental anchor for the explanation of media exposure (Todd and Brody 1980; Mendelsohn and Nadeau 1996; Humanes 2014). Our research reinforces this idea. The results show how emotions towards parties have a greater weight than emotions towards leaders in the choice of media (Hypothesis 3). We observed this specifically through the significant variables in the final models.

The emotional regime of the media is characterized by combining positive and negative emotions. Positive emotions towards leaders and parties are consistent with the media with a close editorial line. Negative emotions are more intense towards leaders and political parties that do not share the editorial line of the media (Hypothesis 1). These emotions also represent the Spanish political context, especially emotional polarization.

These negative emotions appear towards leaders and political parties contrary to the political and ideological line of the media. For example, if someone consumes news that criticizes a particular political party, they are likely to experience negative emotions of anxiety or aversion whenever information is presented that supports that narrative. We understand that the activation of negative aversive emotions or anxiety do not occur in isolation but are part of a complex emotional construction process (Maneiro et al. 2023) This phenomenon can have profound effects on society and political polarization. These results are consistent with previous studies that have dealt with the consumption of information because of emotions linked to anxiety and aversion (Vasilopoulos et al. 2019; Vasilopoulou and Wagner 2017).

Aversive emotions are the most activated ones and the ones that most condition the selection of the media to inform oneself (Hypothesis 2). Emotions of an aversive nature such as hate or disgust are frequent in our models, which describes the political context of emotional polarization in Spain. When people constantly surround themselves with information that validates their political beliefs and emotions (Marcus et al. 2000), they are less likely to be willing to consider other perspectives or find common ground with those who have different opinions. This can increase division in society and hinder dialogue and mutual understanding between groups with divergent political views. In this way, the political fracture and emotional polarization in society appear.

Positive emotions also explain media selection. Positive emotions are significant when they are felt towards a party or political leader close to the editorial line (Hypothesis 1). These results are consistent with previous research associating positive emotions with the

reinforcement of political predispositions and heuristics (Marcus et al. 2000; Lagares et al. 2022a; Mo 2021).

As we have observed in our analysis, media consumption is closely related to the development of a social fracture. Anger is one of the emotions with the greatest presence among media consumers. In addition, our results demonstrate how anger explains media selection.

On the other hand, our exploratory analysis allows us to raise the following discussion, which deserves to be analyzed more precisely in future research. The results suggest that higher newspaper consumption is more closely associated with political emotions, although increased information consumption correlates with lower emotional intensity. These results describe the regime and emotional architecture of newspaper readers. It is a complex emotional regime that combines low intensities of positive and negative emotions. Newspapers are more sophisticated than other media such as radio or television because their audiences are also sophisticated. It is an informed public that looks for specific topics in the newspapers as well as closely aligned and dissonant opinions. Although television reaches many more people, its audience is more dispersed and heterogeneous. These arguments lead us to reinforce the role of newspapers in political campaigns, because they are spaces with very select audiences.

Finally, we consider it important to continue this work by exploring how selective exposure is constructed by taking into consideration emotions and the mediation of partisan identification, as well as ideology, from a multivariate perspective.

**Author Contributions:** Conceptualization, J.M.R.O. and D.M.-G. methodology, J.M.R.O. and D.M.-G.; validation, J.M.R.O., D.M.-G. and G.V.I.; formal analysis, J.M.R.O., D.M.-G. and G.V.I.; investigation J.M.R.O., D.M.-G. and G.V.I.; data curation, J.M.R.O., D.M.-G. and G.V.I.; writing—original draft preparation, J.M.R.O., D.M.-G. and G.V.I., writing—review and editing, J.M.R.O., D.M.-G. and G.V.I.; visualization, J.M.R.O., D.M.-G. and G.V.I.; supervision, J.M.R.O. and D.M.-G.; All authors have read and agreed to the published version of the manuscript.

**Funding:** This research received no external funding.

**Institutional Review Board Statement:** We used data from a non-interventional study and our legislation does not require justification from an ethics committee.

**Informed Consent Statement:** Informed consent was obtained from all subjects involved in the study.

**Data Availability Statement:** The data from the " Estudio Política y Emociones en España, Febrero 2021" are property of the Equipo de Investigaciones Políticas of the University of Santiago de Compostela. The data is not available at this time due to exploitation rights. We used data from a non-interventional study and our legislation does not require justification from an ethics committee. Technical data of the study are provided in the text. If you require more information, please contact the authors.

**Conflicts of Interest:** The authors declare no conflict of interest.

## Appendix A

**Table A1.** Variables introduced in the analysis.

|  | Variable | Type | Interpretation |
|---|---|---|---|
| Emotions | Pride | Nominal (dummy) | 1: Presence of emotion<br>0: Absence of emotion |
|  | Fear | Nominal (dummy) | 1: Presence of emotion<br>0: Absence of emotion |
|  | Hope | Nominal (dummy) | 1: Presence of emotion<br>0: Absence of emotion |
|  | Anxiety | Nominal (dummy) | 1: Presence of emotion<br>0: Absence of emotion |

**Table A1.** *Cont.*

| | Variable | Type | Interpretation |
|---|---|---|---|
| | Enthusiasm | Nominal (dummy) | 1: Presence of emotion<br>0: Absence of emotion |
| | Anger | Nominal (dummy) | 1: Presence of emotion<br>0: Absence of emotion |
| | Hate | Nominal (dummy) | 1: Presence of emotion<br>0: Absence of emotion |
| | Contempt | Nominal (dummy) | 1: Presence of emotion<br>0: Absence of emotion |
| | Worry | Nominal (dummy) | 1: Presence of emotion<br>0: Absence of emotion |
| | Peace of mind | Nominal (dummy) | 1: Presence of emotion<br>0: Absence of emotion |
| | Resentment | Nominal (dummy) | 1: Presence of emotion<br>0: Absence of emotion |
| | Bitterness | Nominal (dummy) | 1: Presence of emotion<br>0: Absence of emotion |
| | Disgust | Nominal (dummy) | 1: Presence of emotion<br>0: Absence of emotion |
| Media consumption frequency | Newspaper, radio, and TV | Ordinal | 6: Every day or almost every day; 5: 4 or 5 days per week; 4: 2 or 3 days per week; 3: Only on weekends, 2: From time to time; 1: Never or almost never |
| Mass media selection | *El País*, *El Mundo*, *ABC*, Antena 3, Telecinco, La Sexta, TVE, La Ser, Onda Cero, COPE | Nominal (dummy) | 1: Consume the media<br>0: Do not consume the media |

Source: authors' own creation based on the data of the "Estudio Política y Emociones en España, Febrero 2021," conducted by the political research team.

**Table A2.** Emotions and tendency to action.

| Emotions | Tendency to Action |
|---|---|
| Enthusiasm | Enthusiasm is a political emotion linked to the reinforcement of political heuristics such as party identification. From a constructivist perspective, enthusiasm builds partisan identification (Jaráiz et al. 2020), learning, political participation, trust. It is built in the present. |
| Pride | Self-exposure, creation of identities, reinforcement of political heuristics, motivation, protest. It is constructed by reference to the past. |
| Hope | Motivation, retainer, goal orientation, reinforcement. It is built by reference to the future. |
| Peace of mind | Control, motivation, pleasant state in the absence of political threats. |
| Hate | Confrontation and conflict, intolerance, elimination of the adversary. Extreme aversive emotion. It is linked to polarization and political fragmentation. Non-reconcilable emotion. |
| Contempt | Denigration, distancing, rejection, discrediting. Extreme aversive emotion. It is linked to polarization and fragmentation. |
| Resentment | Delegitimization, Schadenfreude, construction of identity, confrontation, sustainment of hostilities. Extreme aversive emotion. |
| Bitterness | Relief, social disqualification. Extreme aversive emotion. It is linked to political polarization and fragmentation. |
| Anger | Reduction of threats, reinforcement, repair of transgressions, polarization, mobilization. It is linked to emotional polarization and the political fracture of society. |
| Anxiety | Control of the environment, search for information, interest, polarization, risk aversion, mobilization. |
| Fear | Caution, avoidance, learning, paralysis, authoritarianism, participation inhibition. |
| Disgust | Avoidance, differentiation, rejection, punishment. Extreme aversive emotion. It is linked to polarization and political fragmentation. Non-reconcilable emotion. |
| Worry | Preparedness, resilience. |

Source: own elaboration from Mo (2021).

## Notes

1   For more information, consult Table A2.
2   Tables 9–11 show the values of the odds ratio and the heterocedasticy–robust standard errors.

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
