# Peer review of "Emotions and Media: Emotional Regime and Emotional Factors of Selective Exposure"

_socsci, doi:10.3390/socsci12100554_

Round 1

Reviewer 1 Report

Title: Peer Review Report for "Emotions and Media: Emotional Regime and Emotional Factors of Selective Exposure"

This manuscript presents a compelling exploration of the interplay between emotions and media consumption, emphasizing their significance. However, several areas need improvement to enhance the paper's quality:

Graphical Enhancement of Emotions in Tables (1-6): Incorporating emojis next to each emotion in these tables could provide readers with a more intuitive and visual understanding of emotional states. Furthermore, transforming these tables into heat maps using color codes would offer a clearer representation of emotion levels.

Definition of Emotions: To provide readers with a comprehensive context, it would be beneficial to include clear definitions for each feeling or emotion discussed in the paper. This would help readers grasp the nuances of emotional experiences.

Regarding the inclusion of a Spanish paragraph in the conclusions section, the author should provide an explanation or translation to ensure accessibility for all readers. This oversight could confuse non-Spanish-speaking readers.

Incorporating these suggestions would enhance the visual appeal and comprehensibility of the manuscript, ultimately improving its overall quality and accessibility to a broader audience.

The level of english used on this paper is acceptable, the only thing is that a spanish paragraph was detected in the conclusions section

Author Response

First of all, we would like to thank the reviewer for his valuable comments and suggestions. Regarding these suggestions and recommendations we offer the following responses.

Graphical Enhancement of Emotions in Tables (1-6): We have transformed the tables into heat maps to make emotional results easier to read. On the other hand, we consider that incorporating emojis can be counterintuitive. Some emotions such as bitterness or resentment do not have a defined and intuitive emoji.

Definition of Emotions: Although in the theoretical framework we have alluded to the general behaviors of emotions, we have pointed out and detailed the most important behaviors of each emotion. In addition, we included a summary table in the annex with a brief explanation of the political behaviors associated with each political emotion.

Reviewer 2 Report

You have a nice paper. The introduction goes on, however. too long before at the end you indicate the purpose of your paper. The contribution of the paper should be stated earlier. Although you are evidently not economists, it would enhance the paper to include references to papers from the economics literature. I suggest that you put media bias into google scholar to find a few relevant references. The literature on expressive behavior is also relevant.

Author Response

First, we would like to thank the reviewer for his comments and valuable contributions. We have no doubt that they have improved our article.

We have incorporated the purpose of the article at the beginning. For this we have moved the final paragraph to the beginning. On the other hand, we incorporate some references about media bias and how it can be connected to selective exposure. In addition, we have incorporated the definition of expressive behavior, as well as some references that we consider interesting for our work.

Round 2

Reviewer 1 Report

Thank you for your prompt revisions to the manuscript. I have reviewed the changes, and I am pleased to confirm that I am satisfied with all the corrections made to the paper.